# Reversible Diels–Alder Addition to Fullerenes: A Study of Dimethylanthracene with H_2_@C_60_

**DOI:** 10.3390/nano12101667

**Published:** 2022-05-13

**Authors:** Mahboob Subhani, Jinrong Zhou, Yuguang Sui, Huijing Zou, Michael Frunzi, James Cross, Martin Saunders, Cijun Shuai, Wenjie Liang, Hai Xu

**Affiliations:** 1College of Chemistry and Chemical Engineering, Central South University, South Lushan Road, Changsha 410083, China; mehboobsubhani814@gmail.com (M.S.); 18171099526@163.com (J.Z.); sui_yg@csu.edu.cn (Y.S.); 2Department of Biology, College of Arts and Science, New York University, New York, NY 10012, USA; hz2750@nyu.edu; 3Department of Chemistry, Yale University, New Haven, CT 06520, USA; micheal.frunzi@gmail.com (M.F.); james.cross@yale.edu (J.C.); ms@gaus90.chem.yale.edu (M.S.); 4College of Mechanical and Electrical Engineering, Central South University, South Lushan Road, Changsha 410083, China; 5Shenzhen Research Institute, Central South University, High-Tech Industrial Park, Yuehai Street, Shenzhen 518057, China

**Keywords:** ^1^H NMR spectroscopy, Diels–Alder reaction, 9,10-dimethylanthracene, fullerene, H_2_@C_60_

## Abstract

The study of isolated atoms or molecules inside a fullerene cavity provides a unique environment. It is likely to control the outer carbon cage and study the isolated species when molecules or atoms are trapped inside a fullerene. We report the Diels–Alder addition reaction of 9,10-dimethyl anthracene (DMA) to H_2_@C_60_ while ^1^H NMR spectroscopy is utilized to characterize the Diels–Alder reaction of the DMA with the fullerene. Through ^1^H NMR spectroscopy, a series of isomeric adducts are identified. The obtained peaks are sharp, precise, and straightforward. Moreover, in this paper, H_2_@C_60_ and its isomers are described for the first time.

## 1. Introduction

A fullerene molecule is an allotrope of carbon. Investigating this newly discovered class of form has become a particularly striking and dynamic research area [1,2]. In the 1990s, the formation of an orifice on fullerene C_60_ attracted immense attention for targeting endohedral fullerenes in organic syntheses, such as molecular containers [3,4,5,6,7,8], and investigating their ability as electron acceptors in solar cells. Inside the cavity, endofullerene contains small atoms or molecules, including helium, hydrogen, and water, as well as endohedral metal-fullerenes, such as Li, Th, and Sc_3_N [9,10,11,12,13,14]. Inserting small molecules or atoms into a big closed fullerene cage is an intriguing yet demanding research area for scientists. Moreover, the encased molecule performs like a confined quantum rotor, with a sophisticated energy level structure [15] that could be investigated via neutron scattering, infrared (IR), as well as nuclear magnetic resonance (NMR) spectroscopy [15,16,17,18,19,20,21]. The steadiness and homogeneity of endofullerenes like H_2_@C_60_, HF@C_60_, and H_2_O@C_60_ make it possible to examine crucial physical phenomena like nuclear spin isomer transformation with ease and precision [18,22,23,24,25]. Fullerene’s exterior chemical functionalities are heavily influenced by their bond reactivity.

The Diels–Alder addition reaction is a valuable research area in fullerene chemistry. To functionalize fullerenes, the Diels–Alder reaction is used [26]. When small molecules or atoms are encapsulated in the fullerene, they not only enhance the stability but also increase the Diels–Alder cycloaddition reactivity of the fullerene [27,28,29]. On the one hand, it has been demonstrated that the Diels–Alder reaction of 9,10-dimethylanthracene with C_60_ is reversible at ambient temperature. This phenomenon also happens with C_70_ [30]. On the other hand, a reduction in the exothermicity of the reaction may cause a reversible reaction because two methyl groups can increase steric hindrance. Hirsch has employed the reaction to manipulate the position of several malonate additions by adding removable templates to C_60_ (see Figure 1) [30,31,32]. In an early publication, we successfully employed helium spectroscopy to explain the reactions of C_60_ and C_70_ with DMA and studied the equilibria of the reaction [33]. When helium is inserted inside an endohedral fullerene cage, and its NMR chemical shift is matched with the chemical shift of ^3^He outside, the magnetic field shielded by the fullerene can be measured directly [27,34,35,36,37,38,39,40,41]. This is caused by diamagnetism and is linked to the ring currents in the molecular orbitals of fullerene. Higher fullerenes, He@C_n_, have previously been demonstrated to have helium NMR chemical shifts that fall between two extremes, namely low-field C_60_ and high-field C_70_, indicating that they have an “intermediate” aromatic nature [37,39]. The ^3^He NMR is vital for fullerene chemistry [42,43]. Still, there are some limitations to helium NMR, i.e., the measurement at low temperature demands high sensitivity, low noise, and high pressure. Even if the NMR signal itself is visible, changes in the signals are negligible at the lowest temperature. Besides that, ^3^He can be studied in a specially designed and constructed cryostat. Helium NMR spectra use potassium cyanide, or KCN, which is highly toxic, as well as special ^3^He isotopes, which are very costly. While ^1^H NMR is inexpensive, simple, and easy to operate, the spectra for each adduct give a simple and sharp peak.

In the continuation of our previous work [44], we used the T-Jump Method to the Diels–Alder addition of 9,10-dimethylanthracene to C_60_. It was essential to ensure that the equilibrium data of ^3^He@C_60_ was applied to H_2_@C_60_. Herein, we extended H_2_@C_60_ to investigate the Diels–Alder addition reaction of 9,10-dimethylanthracene and fullerene. Through ^1^H NMR characterization and analysis, we found 1 mono-, 6 bis-, 11 tris-, and 10 tetrakis-adducts.

## 2. Experimental Section

**Preparation of Fullerene** H_2_@C_60_ was prepared by following the previously described method [45,46]. NMR tubes were used to prepare all of the ^1^H NMR samples. A total of 3.6 mg of H_2_@C_60_ was kept in the nuclear magnetic tube. Subsequently, 1 mL of mixed solvent CS_2_: CDCl_3_ = 4:1 was added to dissolve H_2_@C_60_ completely, and then a ^1^H NMR test was performed to find the corresponding spectrum. Firstly, 0.51 mg of DMA (0.5 equiv) was placed in the NMR tube. The reaction mixture (3.6 mg of C_60_ and 1.03 mg of DMA (1 equiv) were mixed in 1 mL of CS_2_: CDCl_3._
^1^H NMR was used to characterize the reaction of C_60_ and DMA. It was found that almost no free DMA could be found by ^1^H NMR ~ 30 min after the mixing) was allowed to stand overnight, and then the corresponding ^1^H NMR test was conducted to reach equilibrium. After that, a weighted amount of dimethyl anthracene was added to the mixture to obtain another sample. To reach a saturated solution, this procedure could be repeated. Increase the mass of DMA in increments of 1.03 mg (1 equiv), 1.54 mg (1.5 equiv), 2.04 mg (2 equiv), 2.57 mg (2.5 equiv), 4.12 mg (4 equiv), 6.19 mg (6 equiv), 10.31 mg (10 equiv), 15.41 mg (15 equiv), and 20.63 mg (20 equiv) until the saturation solution is reached.

**Proton NMR Spectroscopy** A 500 MHz Bruker Avance spectrometer obtained ^1^H NMR spectra. A solvent-gating pulse sequence was adopted to remove a substantial signal from the ring protons on the non-deuterated solvent molecules. Bulky solvent signals diminish the instrument’s automatic gain setting, decreasing sensitivity for smaller ones. A total of 256 scans were acquired for the competitive reaction with a 3-second acquisition time, a 2-second recycling delay, and a 5 Hz line broadening. 

## 3. Results and Discussion

When the DMA concentration is increased at room temperature, the Diels–Alder addition of DMA to C_60_ becomes reversible and yields mono-, bis-, tris-, and tetrakis-adducts, respectively. Greater levels of the higher adducts accumulate as more DMA is taken. A series of mixtures of DMA and H_2_@C_60_ were prepared in 4:1 CS_2_: CDCl_3_. After the establishment of equilibrium, the ^1^H NMR spectra were recorded. When C_60_ reacts with DMA, a new peak in the DMA product peak appears at δ = 8.3, 7.4 ppm, and near δ = 2.3–2.6 ppm in the ^1^H NMR spectrum, although the spectral peaks are complicated and the integral area is relatively small. This is mainly because the addition product of C_60_ and DMA lowers C_60_’s symmetry, and the addition product is an isomer with the same symmetrical features as C_60_, and its distinctive peaks cannot be created in ^1^H NMR. As a result, corresponding to the new spectral peaks one by one using ^1^H NMR is difficult, as illustrated in Figure 2.

In addition, while using H_2_@C_60_ with DMA, the addition product’s ^1^H NMR peak appeared in the high field at δ = −1.39 ppm, which is easier to examine since the corresponding spectral peak is far away from the spectral peak of other functional groups. Adducts with more DMA molecules correspond to the faster-increasing peak if this ratio varies with DMA concentration. Due to the high symmetry of C_60_, there is only one mono-adduct between C_60_ and DMA. When 0.5 equiv DMA reacts with C_60_, its presence can be found by analyzing the ^1^H NMR spectra (as shown in Appendix A Appendix A. Since C_60_ is excessive at this time, the Diels–Alder addition reaction between DMA and C_60_ is dominated by mono-adducts, and mono-adducts account for the majority. Therefore, we speculate that the peak at δ = −4.979 ppm corresponds to the peak of the mono-adduct. The Diels–Alder addition reaction with C_60_ cannot generate the tris- and tetrakis-adducts due to the small amount of DMA, or the concentration of the tris- and tetrakis-adducts is too low to detect. Therefore, we speculate that the peaks at δ = −4.591 ppm and δ = −5.896 ppm are due to a small number of bis-adducts. When the amount of DMA was increased to 1 equiv and 1.5 equiv (as shown in Appendix A Appendix A), the analysis of the ^1^H NMR spectra showed that at this time, the proportion of H_2_@C_60_ was relatively low, that is, the unreacted H_2_@C_60_ continued to react with the newly added DMA, and the intensity of the spectral peak at δ = −4.979 ppm was relatively increased, that is, the proportion of the mono-addition products gradually increased. Meanwhile, 4 new spectral peaks appeared at δ = −5.941, −6.779, −8.276, −8.329 ppm, respectively, which we assumed were bis-adduct peaks because the amount of DMA at this time was only 1 equiv or 1.5 equiv, and it was difficult to generate tris- and tetrakis-adducts in the Diels–Alder addition reaction with C_60_, or the concentration of tris- and tetrakis-adducts was too low to be detected. When the amount of DMA was increased to 2 equiv, the analysis of the ^1^H NMR spectra showed (as shown in Appendix A Appendix A) that the amount of unreacted H_2_@C_60_ continued to decrease and the amount of unreacted and bis-adduct continued to increase. In the meantime, a new spectral peak appeared at δ = −11.733 ppm. The concentration of the tris-adduct should be detected, and we presumed that the new spectral peak is the tris-adduct based on the chemical shift of the bis-adduct and its change trend and that the tris- and tetrakis-adducts are analyzed by this method. A comparative analysis of the spectra of 15 equiv DMA and 20 equiv DMA showed that no new spectral peak appeared except for the increase in the proportion of the tetrakis-adducts (as shown in Appendix A Appendix A) and the relative decrease in the tris-adduct. Therefore, we speculated that the Diels–Alder addition reaction between DMA and H_2_@C_60_ took place without the pentakis- or hextakis-adducts. Or the concentration of the resulting pentakis- or hextakis-adducts may be too low to detect.

As a result, we can estimate the number of DMA addition products based on the ^1^H NMR spectrum peak because the peak intensity ratio of the two peaks is not dependent on DMA concentration. Then they will resemble an equal number of DMA molecules [33], and H_2_@C_60_ will correspond to the adduct utilizing a similar quantity. But when the ratio varies with DMA concentration, faster-rising peaks could be linked to an adduct containing extra DMA molecules. The use simplifies the study of the Diels–Alder addition reaction between C_60_ and DMA. The 29 new ^1^H NMR peaks are evaluated one by one, the fraction of the NMR intensity for each isomer at 298 K, as shown in Figure 3a,c,(a partial enlargement of Figure 3c is shown in Appendix A Appendix A) and the chemical shifts, as well as the ^1^H NMR peaks, are described accordingly.

As shown in Table 1, we found 1 mono-, 6 bis-, 11 tris-, 10 tetrakis-adducts, and an unreacted embedded hydrogen fullerene from the Diels–Alder addition reaction between DMA and C_60_. There were no spectral lines shown with an increasing concentration of DMA. As a result, we did not find any indication of pentakis- and hexakis-adducts in the DMA-C_60_ Diels–Alder addition reaction. Such a conclusion is used by Wang Guanwu et al. at 295.4 K [33] for ^3^He@C_60_, and the obtained peak patterns are very similar, except for the overall shift of the NMR spectrum (as shown in Figure 3b,d). Furthermore, with the increase in DMA concentration, the mono-adduct gradually decreases, and correspondingly, the multi-addition products gradually increase.

The supposed reversible reactions of ^1^H@C_60_ and DMA have been described in the following Equations (1)–(4):(1)$$H_{2}@C_{60}+DMA\overset{K_{1}}{⇄}H_{2}@C_{60}(DMA)$$
(2)$$H_{2}@C_{60}(DMA)+DMA\overset{K_{2}}{⇄}H_{2}@C_{60}{\left(DMA\right)}_{2}$$
(3)$$H_{2}@C_{60}{\left(DMA\right)}_{2}+DMA\overset{K_{3}}{⇄}H_{2}@C_{60}{\left(DMA\right)}_{3}$$
(4)$$H_{2}@C_{60}{\left(DMA\right)}_{3}+DMA\overset{K_{4}}{⇄}H_{2}@C_{60}{\left(DMA\right)}_{4}$$

The calculations of mono-, bis-, tris-, and tetrakis-adducts are measured in these equations. Though the ^1^H spectrum can be used to determine the sums of C_60_ and the mono-, bis-, tris-, and tetrakis-adducts, it cannot determine the quantity of free DMA. As a result, a computer program was written that simulates the experimental results using the concentration of free DMA. A number of values for concentrations of free C_60_ and DMA were obtained for any presumed values of K_1_, K_2_, K_3_, and K_4_. The preceding formula was applied to calculate the total concentrations of the various types of adducts based on these values. When these quantities are added together, the overall concentration of C_60_ is calculated. Similarly, the total volume of DMA was calculated by summing up the number of DMA molecules in each adduct. The experimental results were compared with estimated values. To meet the experimental facts, K_1_, K_2_, K_3_, and K_4_ were modified. The theoretical concentration curves presented in Figure 4 were created using values of 3,608,505.76, 38.89, and 2.42 M^−1^, which best fit the experimental results. As shown, all of the experimental values for C_60_ and C_60_(DMA)_n_ fractions at different DMA/C_60_ ratios are very close to the estimated curves. With the addition of DMA, the concentration of free C_60_ declines until it is close to 0 at 2.5 equiv of DMA, whereas the concentrations of mono-adducts, bis-adducts, and tris-adducts initially increase, reach a maximum value, and then decrease. The concentration of tris-adduct peaks reached the maximum at 6.3 equiv of DMA and then gradually fell. Tetrakis-adduct concentrations continue in an upward direction.

The initial concentrations of H_2_@C_60_ and DMA are 5 mM and 4.74 mM, respectively. From the NMR peak strengths of H_2_@C_60_ and DMA, we can infer the concentrations of each single addition product and multi-addition product. In contrast, the concentration of unreacted DMA can be calculated by subtracting the reacted part from the initial concentration of DMA. According to the spectrum analysis, when the DMA concentration is increased to 2 equiv, the mono-adduct concentration reaches its maximum, and essentially no bis-adduct is formed. As a result, the following formula can be used to derive the corresponding equilibrium constant K_1_:$$K_{1}=\frac{\left[\mathrm{Mono}\right]}{\left[\mathrm{DMA}][H2@C_{60}\right]}$$

The concentrations of the three after the reaction can be calculated using the NMR peak intensities of H_2_@C_60_ and DMA, which are [Mono] = 2.15 mM, [DMA] = 0.1453 mM, and [H_2_@C_60_] = 4.1 mM, respectively.
$$K_{1}=\frac{2.15}{0.1456\times 4.1}=3608{Μ}^{-1}$$

When the concentration of DMA is increased to 4 equiv, the concentration of unreacted H_2_@C_60_ tends to zero, the concentration of mono- and bis-adducts increases and gradually decreases after reaching the maximum value, and the corresponding equilibrium constant K_2_ is derived.
$$K_{2}=\frac{\left[\mathrm{Bis}\right]}{\left[\mathrm{Mono}][\mathrm{DMA}\right]}$$

In the meantime, there are six bis-adducts, so [Bis]_total_ = [Bis]_1_ + [Bis]_2_ + [Bis]_3_ + [Bis]_4_ + [Bis]_5_ + [Bis]_6_, since the amounts of [Bis]_1_ and [Bis]_3_ are very small, so neglected. By analysing the nuclear magnetic peak strength in the spectra, it is obtained that [DMA] = 0.000145 mM, [Bis]_2_ = 0.0108 mM, [Bis]_4_ = 0.0140 mM, [Bis]_5_ = 0.728 mM, [Bis]_6_ = 0.0605 mM, so [Bis]_total_ = 0.158 mM, from which the equilibrium constant K_2_ is calculated.
$$K_{2}=\frac{0.158}{2.15\times 0.000145}=505.76{Μ}^{-1}$$

And so, K_3_ = 38.89 M^−1^ and K_4_ = 2.42 M^−1^.

In 2009 [44], we found very similar results by analyzing the reaction rates and equilibrium constants of DMA and C_60_ using H_2_@C_60_ and ^3^He@C_60_. The chemical shifts of the spectral peaks for the mono-adduct of H_2_@C_60_ and ^3^He@C_60_ are δ = −1.39 ppm, δ = −6.32 ppm, and δ = −4.95 ppm, δ = −9.86 ppm, respectively. Hence, the single mono-adduct shift differences are 3.56 ppm and 3.54 ppm, which are very close, indicating that the embedded H_2_ and ^3^He have little effect on the C_60_ shell reaction (as shown in Figure 5), while in ^129^Xe@C_60,_ the chemical shift between the mono-adduct and unreacted C_60_ is δ = +10.9 ppm [47]. In both scenarios, it is concluded that the endohedral molecule is bouncing around inside the C_60_ cage without creating any substantial interference to the C_60_’s electrons. The carbon cage of fullerene and the molecule inside have a weak van der Waals force, as well as a repulsive force when the molecule approaches the cage. The equilibrium constants of both endohedral fullerenes can then be safely assumed to be the same as those of empty C_60_.

Nonetheless, the synthesis of H_2_@C_60_ is a complex process that takes eight steps to be complete [37,39,48]. H_2_@C_60_ is still preferred to ^3^He@C_60_ for different reasons. (1) Because the matching peaks are far away from the peaks of other functional groups, ^1^H is an excellent NMR probe to analyze H_2_@C_60_ precisely. (2) We can make H_2_@C_60_ with a 100% incorporation ratio, whereas ^3^He@C_60_ has approximately 1%. Since H_2_@C_60_ has two hydrogen atoms versus one helium atom in ^3^He@C_60_, hence giving us a sensitivity boost of more than two orders of magnitude. (3) In proton NMR, there is no issue of noise as compared to helium NMR, as shown in Figure 5. As the endohedral hydrogen resonances occur far up-field from the solvent, the peak separation is easy and well separated from the other remaining spectrum. Furthermore, the force between H_2_ and C_60_ in endohedral fullerenes is a van der Waals force. As a result, the H_2_ atom has little effect on the basic chemical reactivities of C_60_.

## 4. Conclusions

In summary, we have employed ^1^H NMR spectroscopy to explain the reaction between fullerene C_60_ and DMA. Through obtaining several ^1^H NMR spectra of the reaction mixture of H_2_@C_60_ and DMA, we were able to identify a series of DMA molecules that reacted to the fullerene and the isomers that existed in the reaction mixture for each peak in the ^1^H NMR spectroscopy. We have successfully obtained 1 mono-adduct, 6 bis-adducts, 11 tris-adducts, and 10 tetrakis-adducts from C_60_ and DMA. The findings suggest that changing the skeleton of fullerene can substantially impact the cage’s reactivity and that H_2_@C_60_ will be helpful in a number of complex fullerene reactions, such as the conversion of electrophile to nucleophile [49], and is expected to have numerous applications in materials science and technology [50], as well as H2@C60, paves the way to the pyrrolidinoendofullerene [51,52,53]. Further work is underway to investigate H_2_@C_60_ inside the fullerene cage and explore its chemical and physical properties.

## Figures and Tables

**Figure 1 nanomaterials-12-01667-f001:**
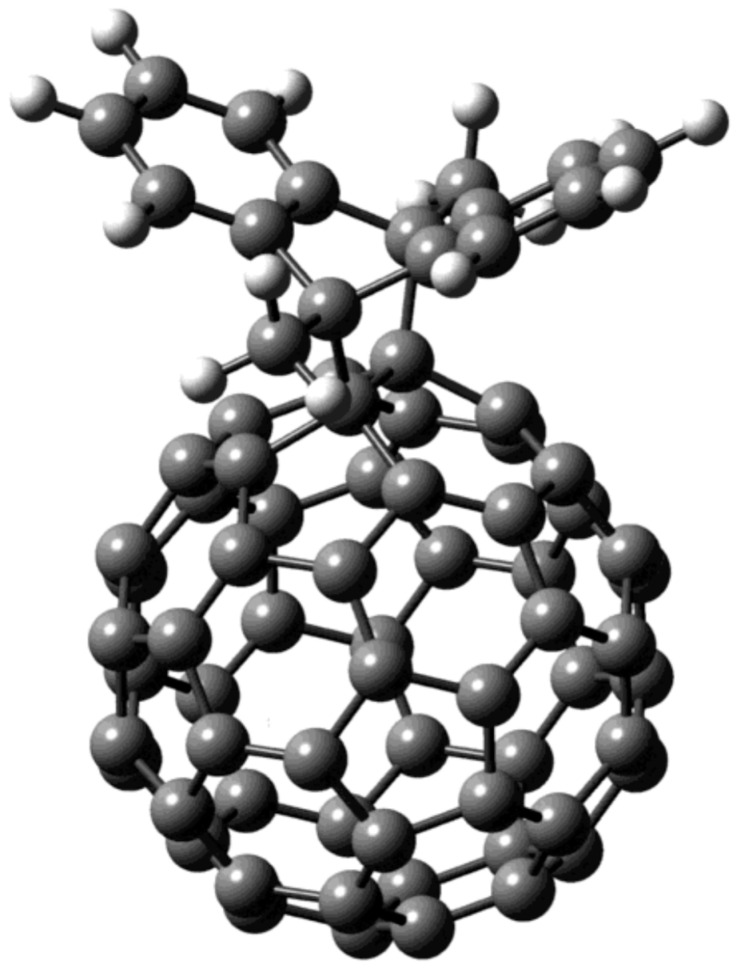
Gauss view of mono-adduct of DMA and C_60_.

**Figure 2 nanomaterials-12-01667-f002:**
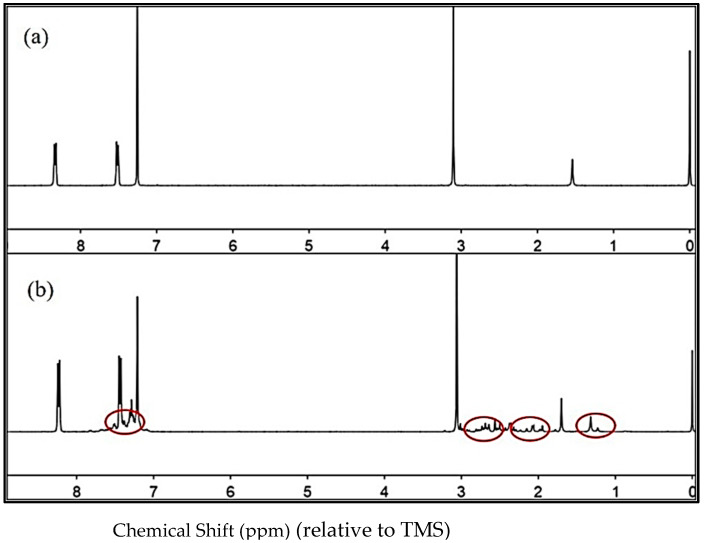
^1^H NMR spectra of (**a**) DMA, (**b**) 1.0 equivalent DMA + C_60_.

**Figure 3 nanomaterials-12-01667-f003:**
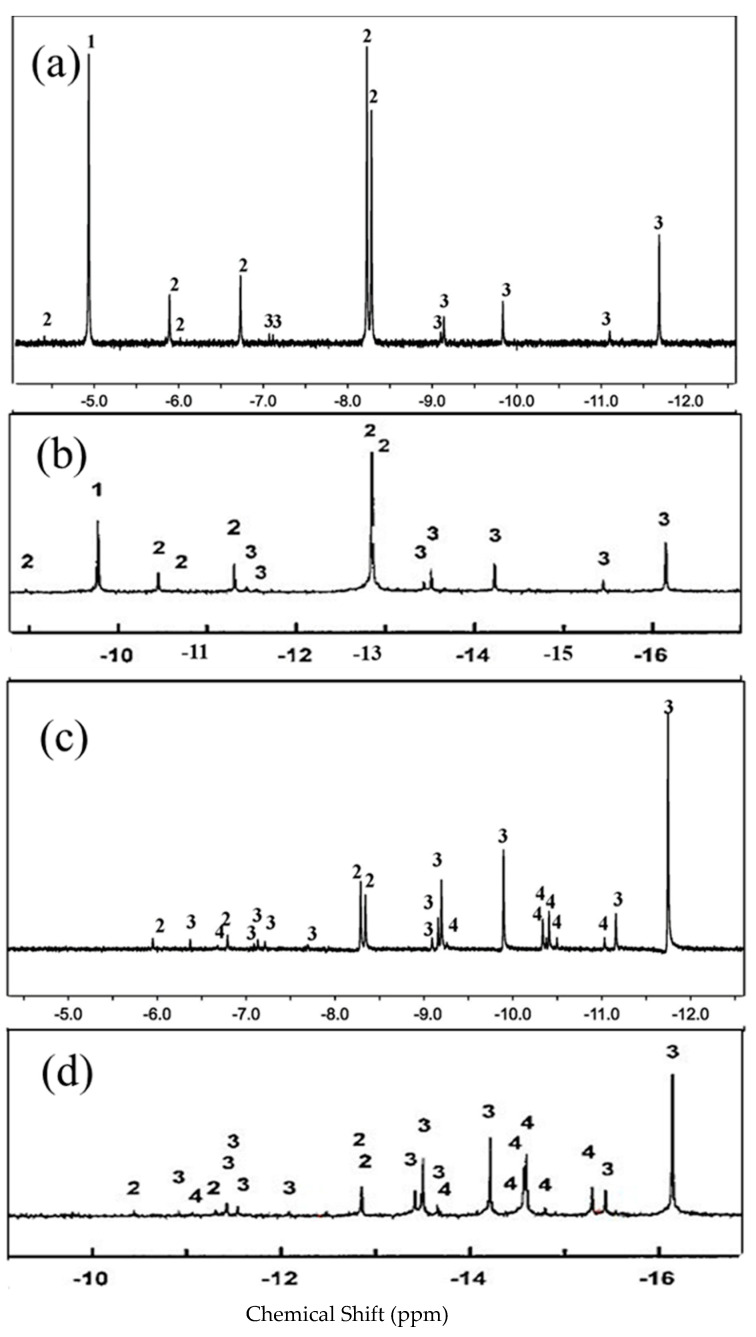
(**a**) ^1^H NMR spectra of H_2_@C_60_ with 2.5 equiv of DMA, (**b**) ^3^He NMR spectra of ^3^He@C_60_ with 2.5 equiv DMA; (**c**) ^1^H NMR spectra of H_2_@C_60_ with 10 equiv of DMA; (**d**) ^3^He NMR spectra of ^3^He@C_60_ with 10 equiv DMA at room temperature. The numbers **1**, **2**, **3**, **4** stand for the isomers of mono-, bis-, tris-, and tetrakis-adducts, respectively. Chemical shift in ppm relative to TMS (for (**a**,**c**)) and dissolved ^3^He gas (for (**b**,**d**) (*J. Am. Chem. Soc.*
**2001**, *123*, 256–259)).

**Figure 4 nanomaterials-12-01667-f004:**
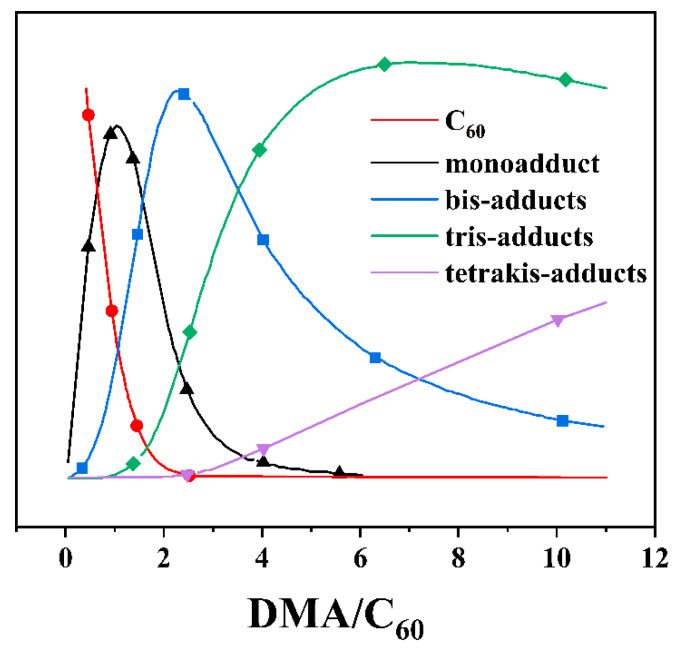
Distribution of C_60_ and C_60_(DMA)_n_ at different ratios of DMA/C_60_.

**Figure 5 nanomaterials-12-01667-f005:**
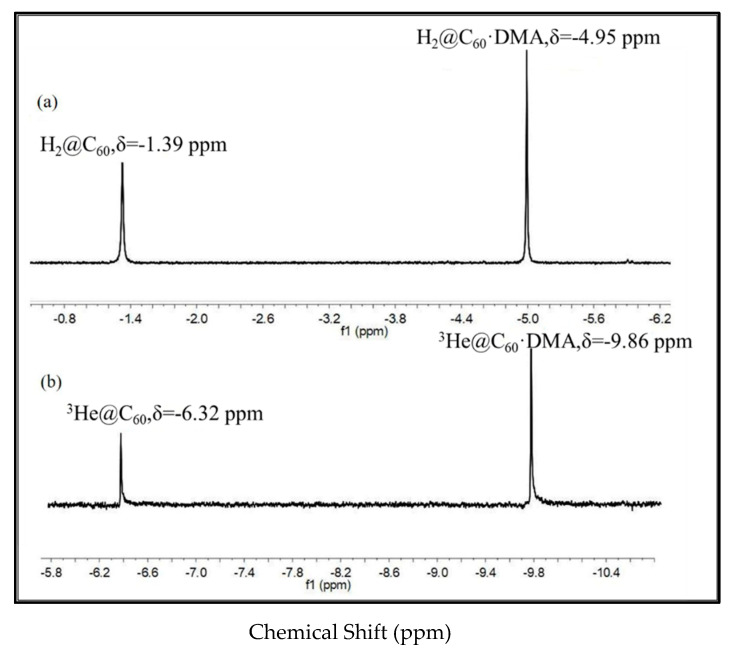
(**a**) ^3^He NMR spectrum of ^3^He@C_60_ and mono-adduct with DMA; (**b**) ^1^H NMR spectrum of H_2_@C_60_ and mono-adduct with DMA [44].

**Table 1 nanomaterials-12-01667-t001:** Chemical shifts of addition products of C_60_ and DMA.

	δ ^a^ (ppm)		f ^b^		δ ^a^ (ppm)		f ^b^
		Mono				Tetrakis	
1	−4.954		1.000	1	−6.667		0.008
		Bis		2	−8.092		0.009
1	−4.591		0.012	3	−9.397		0.031
2	−5.896		0.073	4 5	−10.041 −9.531		0.047 0.031
3	−5.941		0.008	6	−10.321		0.282
4	−6.779		0.092	7	−10.391		0.354
5	−8.276		0.435	8	−10.482		0.042
6	−8.329		0.380	9	−11.013		0.166
		Tris		10	−11.145		0.030
1	−6.073		0.005				
2	−7.076		0.008				
3	−7.120		0.024				
4	−7.198		0.042				
5	−8.618		0.053				
6	−9.142		0.009				
7	−9.182		0.162				
8	−9.249		0.018				
9	−9.881		0.217				
10	−11.142		0.054				
11	−11.733		0.408				

^a^ Chemical shift in ppm relative to TMS; ^b^ Fraction in each isomer of the total NMR signal for all isomers with a given number of DMA addends.

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
