# Peer review of "Reversible Diels–Alder Addition to Fullerenes: A Study of Dimethylanthracene with H2@C60"

_nanomaterials, 2022, doi:10.3390/nano12101667_

Round 1

Reviewer 1 Report

I do appreciate the efforts of the authors.

Unambiguously, the revised manuscript is of better quality and improved the presentation of the main findings of this study.

My comments were mostly addressed by the authors, however a main concern needs to be answered. In my report I mentioned that: "In my honest opinion, Figure 3 is indeed indicative of the presence of multi-adducts of H2@C60 in the reaction mixture, nevertheless the methodology for distinguishing the bis-, tris- and tetrakis-adducts of H2@C60 with DMA is totally unclear. The citation of previous work is not sufficient and the related discussion is not well-written. Further, some of the mentioned peaks in the H2@C60 isomeric 1H NMR spectra are indistinguishable from the baseline signal. All spectra should be better presented in order to assist the reader.". In the revised manuscript this major issue remains unresolved. How did the authors assigned the multi-adducts? Further, the image resolution of Figure 3 remains insufficient, although it was magnified. The authors could transfer Figure 3 in the SI and incorporate a new Figure 3 using some insets to show the peaks originating from the multi-adducts of lower concentrations. 

Finally, in lines 144-145, authors mention that: "As a result, a computer program was created that simulates the experimental results using the concentration of free DMA.". More details should be provided, at least in the experimental section. How this program operates, which are the variables etc.

Author Response

Point-by-point response letter to reviewers’ comments

We are grateful to every one of the reviewers for taking the time to read our work thoroughly. All of the reviewers' comments have been carefully considered and answered in the order listed below. The corresponding changes were also indicated in red font in the revised paper. Thank you so much for all of your assistance!

Reviewer 1

I do appreciate the efforts of the authors. Unambiguously, the revised manuscript is of better quality and improved the presentation of the main findings of this study. My comments were mostly addressed by the authors, however a main concern needs to be answered.

Comment 1: In my report I mentioned that: "In my honest opinion, Figure 3 is indeed indicative of the presence of multi-adducts of H2@C60 in the reaction mixture, nevertheless the methodology for distinguishing the bis-, tris- and tetrakis-adducts of H2@C60 with DMA is totally unclear. The citation of previous work is not sufficient and the related discussion is not well-written. Further, some of the mentioned peaks in the H2@C60 isomeric 1H NMR spectra are indistinguishable from the baseline signal. All spectra should be better presented in order to assist the reader.". In the revised manuscript this major issue remains unresolved. How did the authors assigned the multi-adducts? Further, the image resolution of Figure 3 remains insufficient, although it was magnified. The authors could transfer Figure 3 in the SI and incorporate a new Figure 3 using some insets to show the peaks originating from the multi-adducts of lower concentrations.

Response: Thank you for your kind suggestion. We have described in detail how to distinguish the bis-, tris-, and tetrakis-adducts in the text and added the corresponding 1H NMR spectra in the supporting information to facilitate the readers' understanding and analysis. We have redrawn Figure 3 to a higher resolution and zoomed in where the peaks are denser and inserted them in Figure S7.

Comment 2: Finally, in lines 144-145, authors mention that: "As a result, a computer program was created that simulates the experimental results using the concentration of free DMA. More details should be provided, at least in the experimental section. How this program operates, which are the variables etc.

Response: The relative amounts of C60 and the mono-, bis-, tris- and tetrakis-adducts are observable by using the 1H spectra, the amount of free DMA is not directly observable in the 1H spectrum. This is true for the 3He spectra as well. Therefore, a computer program was written that uses the concentration of free DMA as a parameter in simulating the experimental results. There was a slight mistake in the article. We wanted to say that a computer program was written instead of created. We have added relevant descriptions in the experiment section. The distribution of unreacted H2@C60 and generated mono-, bis-, tris-, and tetrakis-adducts under different equivalent DMA was calculated according to the variation of 1H NMR spectrum peaks, and the origin software was used to perform nonlinear fitting, respectively.

Reviewer 2 Report

This is indeed an excellent study, and the authors have greatly improved the writing, although a few incomplete sentences and grammatical mistakes do persist.  I see no reason to delay publication.

One question occurs to me:  Can you rationalize why there are 6 bis, 11 tris, and 10 tetrakis adducts?  Can you model how 2, 3, or 4 but not 5 DMAs can cluster around the C60?

Author Response

Point-by-point response letter to reviewers’ comments

We are grateful to every one of the reviewers for taking the time to read our work thoroughly. All of the reviewers' comments have been carefully considered and answered in the order listed below. The corresponding changes were also indicated in red font in the revised paper. Thank you so much for all of your assistance!

Reviewer: 2

This is indeed an excellent study, and the authors have greatly improved the writing, although a few incomplete sentences and grammatical mistakes do persist.  I see no reason to delay publication.

Comment 1: One question occurs to me: Can you rationalize why there are 6 bis, 11 tris, and 10 tetrakis adducts? Can you model how 2, 3, or 4 but not 5 DMAs can cluster around the C60?

Response: We have described in detail how to distinguish the bis-, tris-, and tetrakis-adducts in the text and added the corresponding 1H NMR spectra in the supporting information to facilitate the readers' understanding and analysis. We analysed the 1H NMR spectra of 15 equivalents of DMA with H2@C60 and the 1H NMR spectra of 20 equivalents of DMA with H2@C60. Comparative analysis found that between the increase in the proportion of tetra-addition products and the relative decrease in the proportion of tri-addition products, no new peaks appear, so we speculate that there may be no formation of penta-addition products and hexa-addition products, or that the concentrations of penta-addition products and hexa-addition products are too low to be detected.

Author Response

Point-by-point response letter to reviewers’ comments

We are grateful to every one of the reviewers for taking the time to read our work thoroughly. All of the reviewers' comments have been carefully considered and answered in the order listed below. The corresponding changes were also indicated in red font in the revised paper. Thank you so much for all of your assistance!

Reviewer: 3

Comment 1: The authors present a manuscript entitled: “Comparison of 3He and 1H NMR Spectroscopy ... etc.”, in keywords is also mentioned “3He spectroscopy” but their study is limited to 1H NMR observation of trapped H2 molecule inside fullerene cage (H2@C60) and the further observation of H2@C60 chemical reactions with dimethylanthracene (DMA). This title is absolutely inadequate for the presented research work.

Response: Thank you for your kind suggestions, and we have changed the title of the article.

Comment 2: I cannot agree with the final conclusions given in Table 2 which suggest the superiority of the analytical application of H2@C60 over 3He@C60 when fullerene chemical reactions are studied by NMR methods. Certainly, 1H NMR is technically much simpler for any application than the 3He NMR method but helium-3 signals are unique and always prove the presence of helium atoms inside 3He@C60. The same is not true for 1H NMR signals because they can also come from any impurities around the observed objects and not necessarily only from aromatic compounds. In such a case is easy for any analytical errors when false 1H NMR signals can be assigned to modified H2@C60 structures. Moreover, the molecular interactions of a spherical 3He atom are much simpler than H2 molecule in C60 and it is important for a more accurate description, e.q. by quantum chemical calculations.

Response: We agree with you. It is true that both H2@C60 and 3He@C60 have unique features and advantages. So, following your suggestion, Table 2 has been changed and the necessary modifications have been made.

Comment 3: Last but not least, the unacceptably substandard descriptions of spectroscopic results in the Experimental Section, Table 1, and in the captions to Fig. 2 and 3. Where are the units and reference standards of chemical shifts? How they were used and measured in NMR experiments? The manuscript must be improved by adding all necessary experimental descriptions and presenting actual research data in a bit more modest way.

Response: Thank you for your kind suggestions, and we have added all necessary experimental descriptions in the experimental section and in the captions to Figures 2 and 3.

Round 2

Reviewer 1 Report

.

Reviewer 3 Report

It can be very helpful for further 1H NMR studies of endohedral fullerenes.

This manuscript is a resubmission of an earlier submission. The following is a list of the peer review reports and author responses from that submission.

Round 1

Reviewer 1 Report

This work compares the 3He and 1H NMR Spectroscopy of the reversible Diels-Alder addition of dimethylanthracene on 3He@C60 and H2@C60 endofullerenes in equilibrium. It is a follow-up of a previous work of some of the authors (J. Am. Chem. Soc. 2001, 123, 256-259; J. Phys. Chem. A 2009, 113, 4996–4999). In the cited previous studies, the reversible Diels-Alder addition of dimethylanthracene on 3He@C60 and the reaction’s equilibria by 3He NMR Spectroscopy (J. Am. Chem. Soc. 2001), as well as the NMR temperature-Jump Method for measuring reaction rates for the reversible Diels-Alder addition of dimethylanthracene on H2@C60 by 1H NMR spectroscopy (J. Phys. Chem. A 2009, 113, 4996–4999), were described.

According to the authors, H2@C60 and its isomeric adduct mixtures are described for the first time by 1HNMR spectroscopy. Investigation of the isomeric mixture revealed 29 1HNMR signals attributed to one mono-, six bis-, eleven tris-, and ten tetrakis-adducts (line 121). Authors should clarify this point since the number of mentioned adducts is 28. In the previous work (J. Am. Chem. Soc. 2001, 123, 256-259) on 3He@C60 one mono-, six bis-, eleven tris-, and ten tetrakis-adducts have been also identified. Thus, the authors’ statement that the number of isomers obtained for H2@C60 is greater than this of 3He@C60 (lines 66-67), seems to be not accurate. Further, in lines 193-194 it mentioned that H2 atom does not affect the basic chemical reactivities of C60. Do the authors have benchmark studies for empty cage C60 with DMA affording the same number/ratio of multi-adducts? In the case that H2@C60 and 3He@C60 result to different amount of multi-adducts, what is the driving force of this difference in reactivity?

In my honest opinion, Figure 3 is indeed indicative of the presence of multi-adducts of H2@C60 in the reaction mixture, nevertheless the methodology for distinguishing the bis-, tris- and tetrakis-adducts of H2@C60 with DMA is totally unclear. The citation of previous work is not sufficient and the related discussion is not well-written. Further, some of the mentioned peaks in the H2@C60 isomeric 1H NMR spectra are indistinguishable from the baseline signal. All spectra should be better presented in order to assist the reader.

Table 2 summarizes some pros and cons for using 1H and 3He spectroscopy for the investigation of such systems. I believe that what actually Table 2 represents is the pros and cons of using H2@C60 and 3He@C60 as scaffolds for the investigation of such reaction dynamics. I would suggest to the authors to consider this aspect.

Some phrases “H2@C60 will be helpful in a variety of complex fullerene reactions” (lines 22-23), “H2@C60 will be helpful in a number of complex fullerene reactions” (line 203) should be put in context and help the reader to understand why these exotic molecules hold potential in unveiling the complex reaction dynamics of the more abundant and commercially available C60.

Other phrases i.e. “Fullerene derivatives are typically challenging to distinguish using only 3He spectroscopy since isomers with the same symmetrical properties will not show noticeable changes.” (lines 18-20) are misleading. It should be revised. 3HeNMR spectroscopy is by definition applicable only to 3He@C60 endofullerene derivative. Further, the authors are not discussing a combined 1H NMR – 3He NMR spectroscopy methodology but a 1H NMR methodology focusing on the endohedral H2.

A final general remark is that the manuscript needs extended editing to improve the reading.

My recommendation is that the authors should address these remarks and resubmit their work.

Reviewer 2 Report

This is a very nice study, cleverly taking advantage of the highly shielded 1H NMR signals of H2@C60 to distinguish its adducts with DMA.  It is necessary to rewrite the manuscript to make sure that the methodology and reasoning are clearly and precisely described.

The English is terrible, beginning with the abstract: "Studying ... gives a unique environment.", "... it is likely to control the outer carbon cage and study the isolated species", "This manuscript has studied ..."

What are "ring protons on the non-deuterated solvent molecules" when the solvent was CDCl3-CS2? What is meant by "A number of DMA and H2@C60 were prepared"?

What do "chemical shifts of the spectral peaks for the mono-adduct of H2@C60 and 3He@C60 at 1.39 ppm, -6.32 ppm, and -4.95 ppm,  -9.86 ppm" represent?

"Figure 4.4a"